

# A retrospective study on the prognostic value of preoperative C-reactive protein to albumin ratio in patients with oral cavity squamous cell carcinoma

Ku-Hao Fang[1], Chia-Hsuan Lai[2], Cheng-Ming Hsu[3], Ethan Huang[3], Ming-Shao Tsai[3], Geng-He Chang[3], Yi-Chan Lee[4] and Yao-Te Tsai[3]

[1] Department of Otorhinolaryngology-Head and Neck Surgery, Chang Gung Memorial Hospital, Taoyuan, Taiwan
[2] Department of Radiation Oncology, Chang Gung Memorial Hospital, Chiayi, Taiwan
[3] Department of Otorhinolaryngology-Head and Neck Surgery, Chang Gung Memorial Hospital, Chiayi, Taiwan
[4] Department of Otorhinolaryngology-Head and Neck Surgery, Chang Gung Memorial Hospital, Keelung, Taiwan

Corresponding author
Yao-Te Tsai, yaote1215@gmail.com

## ABSTRACT

**Background:** Although the C-reactive protein-to-albumin ratio (CAR) can predict poor outcomes in assorted cancers, its prognostic value in oral cavity squamous cell carcinoma (OSCC) remains unclear. We explored the value of preoperative CAR in predicting clinical outcomes in OSCC patients treated with radical surgery.
**Methods:** All the recommended cutoff values were defined analyzing receiver operating characteristic curves or overall survival (OS). Dichotomization was performed on the basis of optimal CAR cutoff, and we compared the clinicopathological features between groups. Kaplan–Meier analysis was also performed to compare OS curves between the two groups. Univariate and multivariate analyses using the Cox proportional hazards model were conducted to find the clinical characteristics that were most closely correlated with disease free survival (DFS) and overall survival (OS). A nomogram incorporated CAR and several clinicopathological factors was established to predict prognosis and its accuracy was evaluated using concordance index (c-index).
**Results:** In this retrospective study, a total of 326 patients with newly diagnosis of OSCC and received primary surgery between 2008 and 2017 were enrolled. Through the executed ROC curve analyses, the optimal CAR cutoff derived was 0.195 (area under the curve = 0.718, $p < 0.001$), with this cutoff exhibiting a discrimination ability superior to that of other inflammation-based prognostic scores after comparing the area under curves. Multivariate analysis demonstrated that CAR ($\geq$0.195/$<$0.195) was associated with OS (hazard ratio 3.614; 95% CI [1.629–8.018]; $p = 0.002$) and DFS (hazard ratio 1.917; 95% CI [1.051–3.863]; $p = 0.029$). Kaplan–Meier analysis and log rank test revealed a significant difference in DFS and OS curves between patients with low CAR ($<$0.195) and those with high CAR ($\geq$0.195; both $p < 0.001$). The c-index of the nomogram based on TNM system alone was 0.684 and could be increased to 0.801 if CAR and other clinicopathological factors were included.

**Conclusions:** Preoperative CAR could constitute an independent prognostic indicator for OS and DFS prediction in OSCC patients treated with curative surgery. The established nomogram that incorporated CAR and prognostic factors might increase the accuracy of prognostic prediction for patients with OSCC.

## INTRODUCTION

Despite the recent decline in betel quid use, oral cavity squamous cell carcinoma (OSCC) incidence has continued to increase in Taiwan (*Liao et al., 2014*); this increase may be explained by the long-term carcinogenic effect engendered by betel quid chewing and by the detrimental effects engendered by alcohol consumption and cigaret smoking (*Adel et al., 2016*). People's cigaret smoking, betel quid chewing, and alcohol consumption not only cause field cancerization but are also correlated significantly with systemic inflammation (*Oliveira, Rodriguez-Artalejo & Lopes, 2010*; *Shafique et al., 2012*; *Zhang et al., 2016*). Some of the available systemic inflammation indices involve patient-related factors; these indices include pretreatment C-reactive protein (CRP) levels, platelet-to-lymphocyte ratio (PLR), neutrophil-to-lymphocyte ratio (NLR), as well as the modified Glasgow prognostic score (mGPS) and can be applied as survival predictors in patients with various cancers (*Brown et al., 2007*; *Crumley et al., 2006*; *Read et al., 2006*), including head and neck cancer (*Takenaka et al., 2018a*, *2018b*). Of these, the pretreatment mGPS, a score combining CRP and serum albumin levels, better predicts cancer survival compared with the peripheral blood cell count-based prognostic scores (e.g., NLR and PLR) (*Dutta et al., 2011*). Recent studies have indicated that CRP-to-albumin ratio (CAR), also created on serum albumin levels and CRP levels, is a valuable prognosticator in various cancers and may provide more accurate prognostic prediction than other indicators (*Ishizuka et al., 2016*). Because patients with OSCC frequently experience malnutrition and inflammation due to eating disability and immunosuppression, CAR may serve as a novel prognostic indicator in OSCC (*Gellrich et al., 2015*). However, the prognostic value achieved by CAR in OSCC requires comprehensive examination, and a few studies with relatively small patient numbers have probed this ratio's prognostic value in OSCC patients (*Park, Kim & Kim, 2016*). Herein, in patients with OSCC treated with curative intent, we examined the prognostic significance of the following preoperative systemic inflammatory indices: CAR, mGPS, PLR, and NLR, highlighting the correlation of CAR with clinicopathological characteristics and treatment outcomes. In addition, a nomogram model incorporating CAR, sex, age, TNM staging, extracapsular nodal extension (ENE), depth of invasion (DOI), and cancer cell differentiation was established to predict the 3- and 5-year overall survival (OS) for patients with OSCC after curative surgical treatment.

## MATERIALS AND METHODS

### Study patients

We retrospectively reviewed the clinical outcomes of patients newly diagnosed with OSCC and who underwent primary curative surgery with or without adjuvant therapy at the Department of Otorhinolaryngology of Chang Gung Memorial Hospital from January 2008 to December 2017.

We subsequently excluded patients with a history of malignancy, synchronous cancer, infection or inflammatory conditions, or autoimmune disorders; those who had received neoadjuvant therapy; and those with missing CRP or albumin data. Finally, 326 patients were enrolled. Their history of cigaret smoking, betel nut chewing, and alcohol consumption was obtained from clinic notes and patient interviews or from the tumor registry. Cigarette smokers were defined as those who had smoked 1 or more cigarets per day for 1 year or longer; betel nut chewers were defined as those who had chewed 2 betel nuts or more daily for at least 1 year; and alcohol drinkers were defined as those who had consumed more than 1 alcoholic beverage per week for more than 6 months. The Institutional Review Board of Chang Gung Memorial Hospital ratified our study protocol (201901573B0), and the requirement for patient's informed consent was waived by the Institutional Review Board. All patients were subjected to routine preoperative workups including blood tests, physical examinations, magnetic resonance imaging or computed tomography of the head and neck, abdominal echography, chest X-ray, nuclear bone scans, and the detailed medical histories of the patients were recorded. Concurrent neck dissection and intraoperative frozen section controls were used for tumor excision per institutional guidelines, and plastic surgeons used local, free, or pedicled flaps for reconstruction of surgical defects. Pathological TNM staging was recorded according to the American Joint Committee on Cancer (AJCC) Cancer Staging Manual Cancer Staging Manual, Eighth Edition. If indicated, postoperative adjuvant therapy based on the institutional guidelines was administered. Briefly, patients with pathological T4 tumors who had positive lymph nodes underwent adjuvant radiotherapy, and patients with any pathological finding including multiple neck lymph node metastases, positive surgical margin, and ENE received adjuvant concurrent chemoradiotherapy within 6 weeks after surgery. A radiation dose of 66 Gy was administered in 2-Gy daily fractions for 5 days each week, and a cisplatin-based regimen was used for the chemotherapy. Patients were followed up bimonthly for the first year after discharge, at 3-month intervals throughout the second year and at 6-month intervals thereafter. At each follow-up visit, physical examination, laboratory testing, and endoscopy were performed. Follow-up imaging with computed tomography or magnetic resonance imaging was performed every 6 months for 2 years and every 12 months thereafter.

### Inflammation-based prognostic scores

To probe the correlation between survival outcomes and systemic inflammatory indices, we performed preoperative blood laboratory tests within 1 week before curative surgery. According to the blood tests and recorded clinical symptoms and signs, severe
infection status was excluded in all patients. The medical staff collected the hematological and biochemistrical parameters during the treatment from patient's charts. Pretreatment biochemistry values of albumin (reference value: 35–55 g/L) and CRP (reference value: <5 mg/L) were measured using biochemistry automated analyzer (Roche Hitachi Cobas 8000, Rotkreuz, Switzerland) during the study period. Hematological results of lymphocyte, neutrophil, hemoglobin, and platelet were measured using the hematology analyzer (Sysmex SE-9000, Kobe, Japan). Preoperative CAR was calculated as follows: CRP level (expressed in mg/L)/albumin level (expressed in g/L). Similarly, we derived the preoperative NLR and PLR as follows: peripheral blood neutrophil count/lymphocyte count and platelet count/lymphocyte count, respectively. Next, mGPS was calculated using previously published methods (*McMillan, 2008*). Patients with both hypoalbuminemia (<35 g/L) and increased CRP levels (>10 mg/L), with one of these variables, and with none of these variables were assigned the scores of 2, 1, and 0, respectively.

## Statistical analysis

By analyzing the receiver operating characteristic (ROC) curves, we determined the statistically optimal cutoff values of prognostic variables on the basis of inflammation, including the CAR, CRP, mGPS, PLR, and NLR. The area under the ROC curve (AUC) was calculated for determining the various indices' discriminatory ability. We used the Spearman test to investigate the correlation between preoperative CRP and albumin levels and examined the normality of distribution of study data by using Kolmogorov–Smirnov test. Patient follow-up was conducted at the outpatient clinic until death or the cutoff date (December 31, 2018). The log-rank test and Kaplan–Meier analysis were performed to evaluate the long-term survival probability in the high and low CAR groups (defined by the optimal CAR cutoff). The clinicopathological features of the two groups were compared using the Mann–Whitney *U* and chi-square tests for continuous and categorical variables, respectively. Cox proportional hazards models based univariate and multivariate analyses were used to identify independent prognostic factors by calculating their hazard ratios (HRs) and the corresponding 95% confidence intervals (CIs). All aforementioned analyses were performed on SPSS version 21.0 (SPSS, Chicago, IL, USA). Statistical significance was indicated by $p < 0.05$. A multivariate nomogram model incorporating sex, age, overall pathological stage, cell differentiation, ENE, DOI, and preoperative CAR was generated as described by *Kao et al. (2018)* by using the "rms" package in R (version 5.1-0; Vanderbilt University, Nashville, TN, USA). Calibration plots were drawn and a concordance index (c-index) of the established nomogram was calculated to assess the predictive accuracy for OS. A c-index of 0.5 denotes the equivalent of random prediction and that of 1.0 indicates perfect prediction (*Harrell, Lee & Mark, 1996*). We calculated the c-index for traditional OS prediction based on TNM staging alone as well as for prediction with the proposed nomogram models with and without CAR.

# RESULTS

## Baseline characteristics

Baseline clinicopathological characteristics as well as laboratory data are listed in Table 1. In total, 326 patients—294 (90.2%) of whom were men and 32 (9.8%) were women—who

**Table 1 Baseline clinicopathological and laboratory characteristics of 326 patients with OSCC.**

| Variable | Characteristics |
| --- | --- |
| Age (years) | |
| <65 | 242 (74.2%) |
| ≥65 | 84 (25.8%) |
| Sex | |
| Men | 294 (90.2%) |
| Women | 32 (9.8%) |
| Primary tumor site | |
| Tongue | 126 (38.7%) |
| Buccal mucosa | 104 (31.9%) |
| Gingiva | 43 (13.2%) |
| Retromolar trigone | 20 (6.1%) |
| Lip | 14 (4.3%) |
| Mouth floor | 13 (4.0%) |
| Hard palate | 6 (1.8%) |
| Cigarette smoking | 267 (81.9%) |
| Alcohol consumption | 215 (66.0%) |
| Betel nut chewing | 260 (79.8%) |
| TNM staging | |
| I | 71 (21.7%) |
| II | 64 (19.6%) |
| III | 39 (11.9%) |
| IV | 152 (46.6%) |
| pT classification | |
| T1 | 90 (27.6%) |
| T2 | 96 (29.4%) |
| T3 | 22 (6.7%) |
| T4 | 118 (36.2%) |
| Nodal status | |
| Metastasis (−), ENE (−) | 209 (64.1%) |
| Metastasis (+), ENE (−) | 51 (15.6%) |
| Metastasis (+), ENE (+) | 66 (20.2%) |
| Cell differentiation | |
| Well | 94 (28.8%) |
| Moderate | 194 (59.5%) |
| Poor | 38 (11.7%) |
| Depth of invasion ≥ 10 mm | |
| Yes | 153 (46.9%) |
| No | 173 (53.1%) |
| Adjuvant therapy | |
| Absent | 178 (54.6%) |
| RT | 43 (13.2%) |
| CCRT | 105 (32.2%) |

(Continued)

| Table 1 (continued) | |
| --- | --- |
| **Variable** | **Characteristics** |
| mGPS | |
| 0 | 227 (69.6%) |
| 1 or 2 | 99 (30.4%) |
| CAR, median (IQR) | 0.08 (0.03–0.34) |
| NLR, median (IQR) | 2.37 (1.73–3.42) |
| PLR, median (IQR) | 114.01 (87.60–154.00) |

**Note:**

OSCC, oral cavity squamous cell carcinoma; ENE, extracapsular nodal extension; RT, radiotherapy; CCRT, concurrent chemoradiotherapy; mGPS, modified Glasgow prognostic score; CAR, C-reactive protein-to-albumin ratio; NLR, neutrophil-to-lymphocyte ratio; PLR, platelet-to-lymphocyte ratio; SD, standard deviation.

were newly diagnosed as having OSCC and underwent primary radical surgery were included; their median age and follow-up duration were 57 (range, 31–86) years and 48 (range, 3–115) months, respectively. The most common primary tumor site was the tongue ($n = 126$, 38.7%), followed by the buccal area ($n = 104$, 31.9%) and gingiva ($n = 43$, 13.2%). Of these patients, 81.9% were smokers, 79.8% were betel nut chewers, and 66% were alcohol consumers. Nearly half of the patients ($n = 152$, 46.6%) were diagnosed as having stage IV disease, and 117 (35.8%) patients had pathologically confirmed neck lymph node metastasis. All the enrolled patients completed the planned treatment course, with 43 (13.2%) patients undergoing only adjuvant radiotherapy and 105 (32.2%) receiving adjuvant concurrent chemoradiotherapy.

## Inflammation-based prognostic score cutoffs and ROC curves

The median CAR was 0.08 (range, 0.01–5.48). A Spearman test revealed that CRP levels were negatively correlated with albumin levels ($r = -0.223$; $p < 0.001$, Fig. 1). By analyzing the ROC curves, we determined the optimal OS cutoff value to be 0.195 for CAR (sensitivity, 65.3%; specificity, 78.4%), 4.505 for NLR, and 165.85 for PLR. We further compared the AUCs of various indices (Fig. 2) to assess their discrimination ability and found that the AUC of CAR (0.718, 95% CI [0.654–0.782], $p < 0.001$) was higher than that of NLR (0.621, 95% CI [0.550–0.692], $p = 0.001$), CRP (0.705, 95% CI [0.638–0.766], $p = 0.001$), PLR (0.610, 95% CI [0.539–0.680], $p = 0.002$), and mGPS (0.679, 95% CI [0.612–0.746], $p < 0.001$).

## Association of CAR with clinicopathological characteristics

We performed dichotomization of patients by the optimal cutoff of CAR, followed by comparing the two groups' clinicopathological characteristics (Table 2). The CAR $\geq$ 0.195 group was determined to have a significant association with advanced TNM stage ($p < 0.001$), lymph node metastasis with extracapsular nodal extension (ENE, $p < 0.001$), DOI of >10 mm ($p < 0.001$), need for adjuvant therapy ($p = 0.003$), higher mGPS ($p < 0.001$), higher NLR ($p < 0.001$), higher PLR ($p = 0.034$), and shortened survival period ($p = 0.01$) compared with the other group.

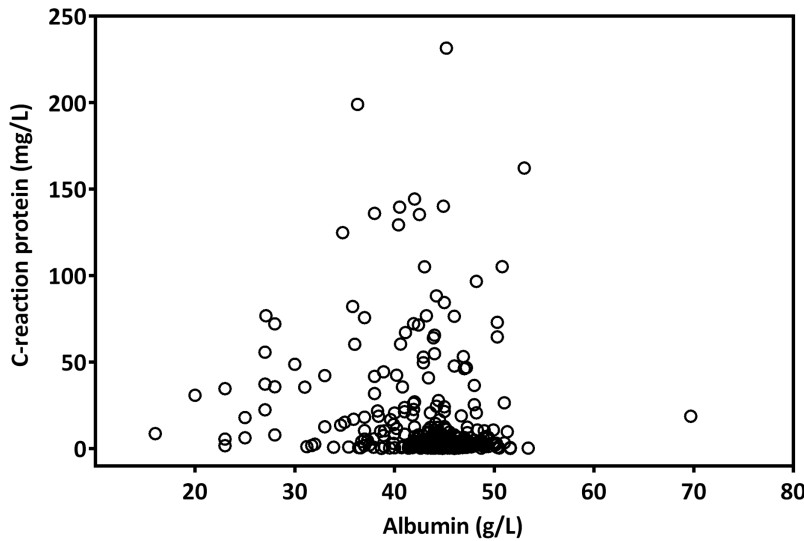

**Figure 1 The scatter plot of the correlation between pretreatment CRP and albumin levels in patients with OSCC.**

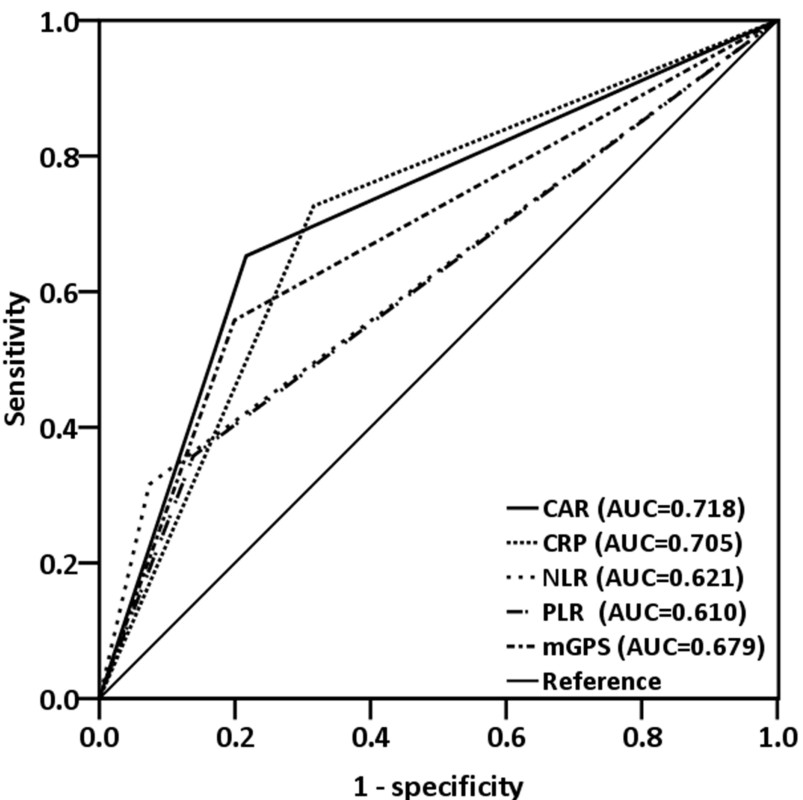

**Figure 2 ROC curves were applied to compare predictive ability of four inflammation-based prognostic scores.** The AUC for CAR was larger (0.718) than that for CRP (0.705), mGPS (0.679), NLR (0.621), and PLR (0.610).

**Table 2 Baseline clinicopathological characteristics according to the CAR.**

| Variable | Number of patients | | p Value |
|---|---|---|---|
| | CAR < 0.195 (n = 214) | CAR ≥ 0.195 (n =112) | |
| Sex | | | 0.434* |
| Men | 191 (89.3%) | 103 (92.0%) | |
| Women | 23 (10.7%) | 9 (8.0%) | |
| Age | | | 0.170* |
| <65 | 164 (76.6%) | 78 (69.6%) | |
| ≥65 | 50 (23.4%) | 34 (30.4%) | |
| TNM staging | | | <0.001* |
| I | 57 (26.6%) | 14 (12.5%) | |
| II | 51 (23.8%) | 13 (11.6%) | |
| III | 24 (11.2%) | 15 (13.4%) | |
| IV | 82 (38.4%) | 70 (62.5%) | |
| pT classification | | | <0.001* |
| T1 | 71 (33.2%) | 19 (17.0%) | |
| T2 | 69 (32.2%) | 27 (24.1%) | |
| T3 | 11 (5.1%) | 11 (9.8%) | |
| T4 | 63 (29.4%) | 55 (49.1%) | |
| Nodal status | | | <0.001* |
| Metastasis (−), ENE (−) | 154 (72.0%) | 55 (49.1%) | |
| Metastasis (+), ENE (−) | 31 (14.5%) | 20 (17.9%) | |
| Metastasis (+), ENE (+) | 29 (13.6%) | 37 (33.0%) | |
| Cell differentiation | | | 0.205* |
| Well | 59 (27.6%) | 35 (31.3%) | |
| Moderate | 134 (62.6%) | 60 (53.6%) | |
| Poor | 21 (9.8%) | 17 (15.2%) | |
| Depth of invasion ≥ 10 mm | | | <0.001* |
| No | 134 (62.6%) | 39 (34.8%) | |
| Yes | 80 (37.4%) | 73 (65.2%) | |
| Adjuvant therapy | | | 0.003* |
| Absent | 130 (60.7%) | 48 (42.9%) | |
| RT | 28 (13.1%) | 15 (13.4%) | |
| CCRT | 56 (26.2%) | 49 (43.8%) | |
| mGPS | | | <0.001* |
| 0 | 214 (100.0%) | 13 (11.6%) | |
| 1 or 2 | 0 (0%) | 99 (88.4%) | |
| NLR (mean ± SD) | 2.5 ± 1.4 | 3.5 ± 2.2 | <0.001** |
| PLR (mean ± SD) | 119.5 ± 52.2 | 149.8 ± 99.8 | 0.005** |
| Survival in months, mean (95% CI) | 48.5 [44.8–52.1] | 39.2 [33.2–45.1] | 0.001** |

**Note:**
CAR, C-reactive protein-to-albumin ratio; ENE, extracapsular nodal extension; RT, radiotherapy; CCRT, concurrent chemoradiotherapy; mGPS, modified Glasgow prognostic score; NLR, neutrophil-to-lymphocyte ratio; PLR, platelet-to-lymphocyte ratio; SD, standard deviation; CI, confidence interval * the Chi-square test ** the Mann–Whitney $U$ test (Z-test: NLR: −4.65; PLR: −2.81; Survival in months: −3.28).

## Association of CAR with survival outcomes

In the univariate analysis, the indicators of poor OS were found to be increased T classification, lymph node metastasis with ENE, poor cell differentiation, DOI >10 mm, need for adjuvant chemoradiotherapy, mGPS of 1 or 2, CAR of ≥0.195, NLR of ≥4.505, and PLR of ≥165.85 (Table 3). Furthermore, multivariate analysis indicated increased T classification ($p = 0.025$ for T3 and 0.003 for T4), ENE ($p < 0.001$), poor cell differentiation ($p < 0.001$), CAR of ≥0.195 ($p = 0.002$), and NLR of ≥4.505 ($p = 0.006$) to be independent prognostic factors for poor OS (Table 3). In the OS probability analysis, the observed 5-year OS incidence was 80.9% and 46.5% in patients with an optimal CAR cutoff of <0.195 and ≥0.195, respectively; these survival differences were significant according to the executed log-rank test ($p < 0.001$, Fig. 3A). The median OS times for patients with CARs of ≥0.195 and <0.195 were 40 (95% CI [12–68]) and >99 months, respectively.

Table 4 demonstrates the association of clinicopathological variables with 5-year DFS. In the univariate analysis, T4 classification, ENE, poor cell differentiation, DOI of >10 mm, need for adjuvant chemoradiotherapy, mGPS of 1 or 2, CAR of ≥0.195, NLR of ≥4.505, and PLR of ≥165.85 were significantly associated with poor DFS. Multivariate analysis results indicated that T4 classification ($p = 0.031$), ENE ($p < 0.001$), poor cell differentiation ($p = 0.009$), CAR of ≥0.195 ($p = 0.029$), and NLR of ≥4.505 ($p = 0.02$) were independent prognostic indicators of poor DFS. In the DFS probability analysis, the 5-year DFS incidence for patients stratified into the CAR < 0.195 and CAR ≥ 0.195 subgroups was at a proportion of 60.1% and 36.8%, respectively; moreover, the log-rank test results demonstrated these differences in DFS to be significant ($p < 0.001$; Fig. 3B). The median DFS times for patients with CARs of ≥0.195 and <0.195 were 24 (95% CI [15–33]) and 86 (95% CI [62–110]) months, respectively.

## Discrimination ability of CAR in subgroup analysis

CAR was significantly correlated with OS in the subgroups of patients with early- and late-stage disease (HR = 7.06, 95% CI [2.58–19.29], $p < 0.001$; HR = 2.97, 95% CI [1.86–4.75], $p < 0.001$, respectively. Fig. 4). Nevertheless, CAR was not significantly associated with OS in the subgroup of patients with lymph node metastasis without ENE.

## Nomogram models

To improve OSCC survival prediction, we established a multivariate nomogram model consisting of CAR, TNM stage, and several clinicopathological factors (Fig. 5A). Figure 5B presents nomogram calibration plots for 3-year OS prediction, and Fig. 5C shows the calibration plots for predicting 5-year OS probabilities. The c-index was 0.801 for the nomogram incorporating CAR and clinicopathological prognosticators, higher than that of the nomograms consisting of clinical factors without CAR (0.759) or TNM staging alone (0.685).

## DISCUSSION

CAR is a strong prognostic indicator for various cancer types. Our literature review revealed that our patient series is the largest thus far and that our study is the first to

**Table 3 Univariate and multivariate analysis of poor prognostic factors for OS in OSCC patients.**

| Variable | 5-year OS (%) | Univariate analysis | | Multivariate analysis | |
|---|---|---|---|---|---|
| | | HR (95% CI) | *p* Value | HR (95% CI) | *p* Value |
| Sex | | | | | |
| Women | 77.3 | Reference | | Reference | |
| Men | 67.7 | 1.595 [0.737–3.452] | 0.236 | 1.069 [0.455–2.516] | 0.878 |
| Age (years) | | | | | |
| < 65 | 70.1 | Reference | | Reference | |
| ≥ 65 | 64.6 | 1.332 [0.864–2.054] | 0.194 | 1.543 [0.926–2.569] | 0.096 |
| pT classification | | | | | |
| T1 | 87.5 | Reference | | Reference | |
| T2 | 73.0 | 1.893 [0.964–3.718] | 0.064 | 2.182 [0.912–4.681] | 0.073 |
| T3 | 62.9 | 3.177 [1.316–7.672] | 0.010 | 3.206 [1.158–8.875] | 0.025 |
| T4 | 51.1 | 3.998 [2.165–7.385] | <0.001 | 3.361 [1.513–7.465] | 0.003 |
| Nodal status | | | | | |
| Metastasis (−), ENE (−) | 79.7 | Reference | | Reference | |
| Metastasis (+), ENE (−) | 60.8 | 2.033 [1.161–3.560] | 0.013 | 1.445 [0.796–2.623] | 0.226 |
| Metastasis (+), ENE (+) | 40.4 | 4.405 [2.813–6.899] | <0.001 | 2.725 [1.617–4.593] | <0.001 |
| Cell differentiation | | | | | |
| Well | 75.7 | Reference | | Reference | |
| Moderate | 71.2 | 1.476 [0.889–2.450] | 0.132 | 1.822 [0.936–3.204] | 0.057 |
| Poor | 40.6 | 3.911 [2.124–7.201] | <0.001 | 4.314 [2.104–8.843] | <0.001 |
| Depth of invasion ≥ 10 mm | | | | | |
| No | 76.7 | Reference | | Reference | |
| Yes | 59.7 | 2.108 [1.396–3.182] | <0.001 | 0.624 [0.346–1.127] | 0. 0.118 |
| Adjuvant therapy | | | | | |
| Absent | 76.5 | Reference | | Reference | |
| RT | 69.9 | 1.571 [0.834–2.962] | 0.162 | 1.577 [0.786–3.163] | 0.199 |
| CCRT | 55.8 | 2.363 [1.529–3.653] | <0.001 | 1.173 [0.677–2.031] | 0.573 |
| mGPS | | | | | |
| 0 | 77.3 | Reference | | Reference | |
| 1 or 2 | 49.0 | 3.491 [2.325–5.241] | <0.001 | 0.916 [0.418–2.007] | 0.827 |
| CAR | | | | | |
| <0.195 | 80.9 | Reference | | Reference | |
| ≥0.195 | 46.5 | 4.397 [2.880–6.714] | <0.001 | 3.614 [1.629–8.018] | 0.002 |
| NLR | | | | | |
| <4.505 | 75.8 | Reference | | Reference | |
| ≥4.505 | 22.2 | 4.515 [2.911–7.003] | <0.001 | 2.271 [1.263–4.085] | 0.006 |
| PLR | | | | | |
| <165.85 | 75.9 | Reference | | Reference | |
| ≥165.85 | 37.8 | 3.338 [2.176–5.122] | <0.001 | 1.576 [0.874–2.843] | 0.131 |

Note:
OS, overall survival; OSCC, oral cavity squamous cell carcinoma; HR, Hazard ratio; CI, confidence interval; ENE, extracapsular nodal extension; RT, radiotherapy; CCRT, concurrent chemoradiotherapy; mGPS, modified Glasgow prognostic score; CAR, C-reactive protein-to-albumin ratio; NLR, neutrophil-to-lymphocyte ratio; PLR, platelet-to-lymphocyte ratio.

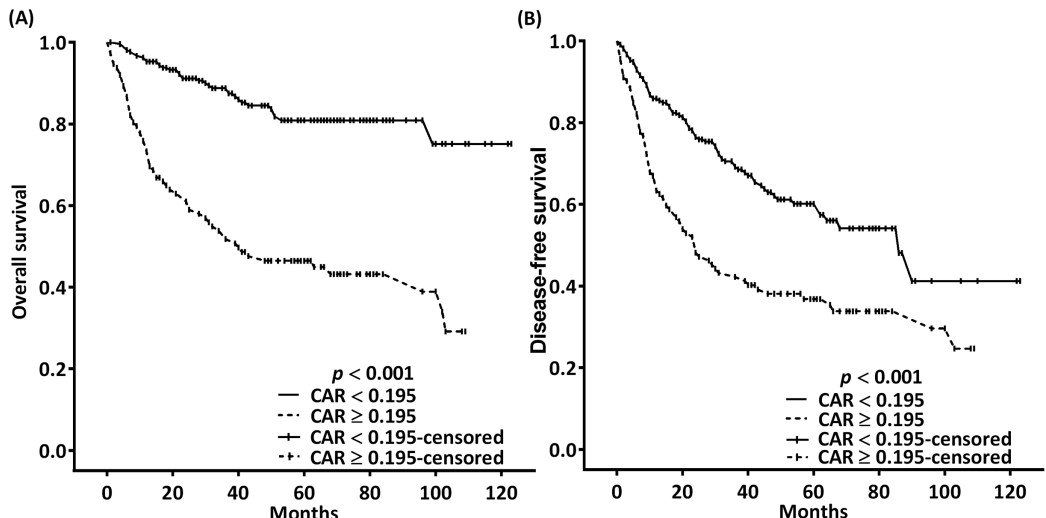

**Figure 3 Kaplan–Meier estimates of 5-year OS (A) and DFS (B) according to the optimal pretreatment CAR cutoffs.**

develop prognostic nomogram model incorporating CAR in patients with OSCC who underwent curative surgery.

By integrating preoperative CAR and diverse prognostic factors, we developed a prognostic nomogram to improve survival prediction for patients with OSCC after surgery. The preoperative nomogram's primary advantage is that it can predict individualized 3- and 5-year survival, thereby helping surgeons to identify patients who are likely to benefit from extensive surgery and multidisciplinary management. Nomograms have been used as prognostic adjuncts for several types of cancer; thus, they play a major role in personalized oncology medicine (*Kao et al., 2018*; *Kattan et al., 1998*; *Li et al., 2016*). In the present study, a high CAR was strongly associated with a more advanced disease stage, lymph node metastasis with ENE, DOI >10 mm, need for adjuvant therapy, and shorter survival. In addition, the multivariate analysis results revealed high CAR as an independent predictor of lower OS and DFS probability, and only CAR and NLR were significant inflammation-based prognostic biomarkers. ROC curve analysis results also suggested that, compared with CRP, mGPS, NLR, and PLR, CAR has superior discriminatory ability for predicting survival in patients with OSCC. In addition, the subgroup analysis of CAR by disease stage confirmed the prognostic value of CAR. Numerous factors affect cancer treatment outcomes. When making decisions in clinical practice, physicians often consider prognostic factors not involved in TNM staging. The majority of such factors are excluded from the TNM staging system because they cannot be used to independently predict survival outcomes in multivariate models. However, many indicators of OSCC with this limitation potentially influence each other; therefore, omitting them from a staging system may reduce the accuracy of survival prediction. Our study established a multivariate nomogram that integrates clinicopathological variables, including CAR, into the conventional TNM staging system and generates individual probabilities of survival outcomes. Compared with that of the

**Table 4 Univariate and multivariate analysis of poor prognostic factors for DFS in OSCC patients.**

| Variable | 5-year DFS (%) | Univariate analysis | | Multivariate analysis | |
|---|---|---|---|---|---|
| | | HR (95% CI) | p Value | HR (95% CI) | p Value |
| Sex | | | | | |
| Women | 49.9 | Reference | | Reference | |
| Men | 68.5 | 1.533 [0.844–2.785] | 0.160 | 1.054 [0.532–2.087] | 0.881 |
| Age (years) | | | | | |
| <65 | 51.6 | Reference | | Reference | |
| ≥65 | 52.9 | 0.935 [0.646–1.352] | 0.720 | 1.041 [0.686–1.580] | 0.849 |
| pT classification | | | | | |
| T1 | 62.8 | Reference | | Reference | |
| T2 | 57.6 | 1.031 [0.647–1.643] | 0.898 | 1.147 [0.695–1.893] | 0.592 |
| T3 | 47.6 | 1.535 [0.777–3.030] | 0.217 | 1.468 [0.677–3.181] | 0.331 |
| T4 | 38.8 | 1.983 [1.311–3.001] | 0.001 | 1.881 [1.061–3.335] | 0.031 |
| Nodal status | | | | | |
| Metastasis (−), ENE (−) | 60.2 | Reference | | Reference | |
| Metastasis (+), ENE (−) | 50.8 | 1.261 [0.791–2.009] | 0.330 | 1.285 [0.779–2.121] | 0.326 |
| Metastasis (+), ENE (+) | 27.1 | 2.713 [1.884–3.907] | <0.001 | 2.279 [1.487–3.494] | <0.001 |
| Cell differentiation | | | | | |
| Well | 49.9 | Reference | | Reference | |
| Moderate | 57.4 | 0.946 [0.656–1.365] | 0.766 | 1.058 [0.710–1.577] | 0.780 |
| Poor | 33.2 | 2.029 [1.240–3.319] | 0.005 | 2.104 [1.206–3.672] | 0.009 |
| Depth of invasion ≥ 10 mm | | | | | |
| No | 55.9 | Reference | | Reference | |
| Yes | 47.6 | 1.379 [0.999–1.902] | 0.050 | 0.808 [0.512–1.277] | 0.361 |
| Adjuvant therapy | | | | | |
| Absent | 55.5 | Reference | | Reference | |
| RT | 55.8 | 1.052 [0.635–1.744] | 0.843 | 0.911 [0.530–1.566] | 0.737 |
| CCRT | 44.4 | 1.416 [1.001–2.003] | 0.049 | 0.745 [0.485–1.143] | 0.178 |
| mGPS | | | | | |
| 0 | 57.5 | Reference | | Reference | |
| 1 or 2 | 39.1 | 1.867 [1.344–2.592] | <0.001 | 0.859 [0.421–1.752] | 0.676 |
| CAR | | | | | |
| <0.195 | 60.1 | Reference | | Reference | |
| ≥0.195 | 36.8 | 2.081 [1.506–2.875] | <0.001 | 1.917 [1.051–3.863] | 0.029 |
| NLR | | | | | |
| <4.505 | 57.5 | Reference | | Reference | |
| ≥4.505 | 16.7 | 2.708 [1.840–3.986] | <0.001 | 1.861 [1.104–3.138] | 0.020 |
| PLR | | | | | |
| <165.85 | 56.6 | Reference | | Reference | |
| ≥165.85 | 32.9 | 2.015 [1.394–2.913] | <0.001 | 1.123 [0.677–1.864] | 0.652 |

Note:
DFS, disease-free survival; OSCC, oral cavity squamous cell carcinoma; HR, Hazard ratio; CI, confidence interval; ENE, extracapsular nodal extension; RT, radiotherapy; CCRT, concurrent chemoradiotherapy; mGPS, modified Glasgow prognostic score; CAR, C-reactive protein-to-albumin ratio; NLR, neutrophil-to-lymphocyte ratio; PLR, platelet-to-lymphocyte ratio.

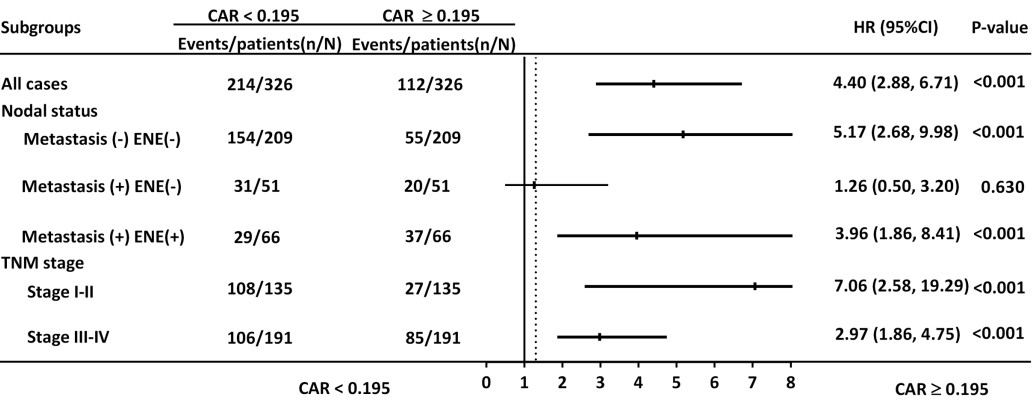

**Figure 4 Hazard ratios (HRs) of CAR in different patient subgroups identified by neck metastatic lymphadenopathy and TNM staging.** HRs > 1.0 indicated a worse outcome.

nomogram excluding CAR (0.759), the c-index of the model with CAR was higher (0.801), suggesting the informativeness of CAR in OSCC survival prediction. The c-indices of both nomograms incorporating clinicopathological factors with and without CAR were higher than that of the nomogram based on TNM staging alone (0.684). The current results support our expectation that preoperative CAR is a valuable biomarker for predicting survival outcomes in patients with OSCC.

Systemic inflammation response possibly has a role in OSCC pathogenesis and progression. OSCC leads to interleukin 6 production, and this potentially promotes CRP synthesis in the liver (*St. John et al., 2004*) and play a role as an autocrine tumor growth factor to enhance oral cancer progression (*Duffy et al., 2008*). Cancer cell invasion and tumor necrosis can positively upregulate systemic inflammatory response, and the inflammatory marker CRP has been previously determined to exhibit an association with survival outcomes in OSCC patients (*Huang et al., 2012*). In addition, as an indicator of chronic malnutrition, hypoalbuminemia was associated with increased wound infection risk and poor prognosis in patients with head and neck cancer (*Danan et al., 2016*). Consistent with previous results (*Hwang et al., 2015*), the current study results indicate a negative correlation between preoperative CRP and albumin levels. This relationship may in part be explained the systemic inflammation-related decreases in albumin levels synthesized in hepatic cells (*Don & Kaysen, 2004*). Systemic inflammation followed by a decrease in serum albumin levels may lead to sarcopenia, nutritional deficiency, and subsequent poor performance. All these factors could have adverse effects on head and neck cancer prognosis (*Bano et al., 2017*).

Various prognostic prediction models, evaluated by peripheral blood cell counts and systemic inflammatory mediators, have been developed to stratify OSCC patients for optimal treatment. In OSCC patients, mGPS is a potential independent prognostic factor of cancer-specific survival and OS (*Farhan-Alanie, McMahon & McMillan, 2015*), and preoperative circulating CRP levels are associated with pathological aggression and survival outcomes (*Chen et al., 2011*). Recent meta-analyses have concluded that elevated pretreatment NLR and PLR demonstrated an association with poor prognosis in patients

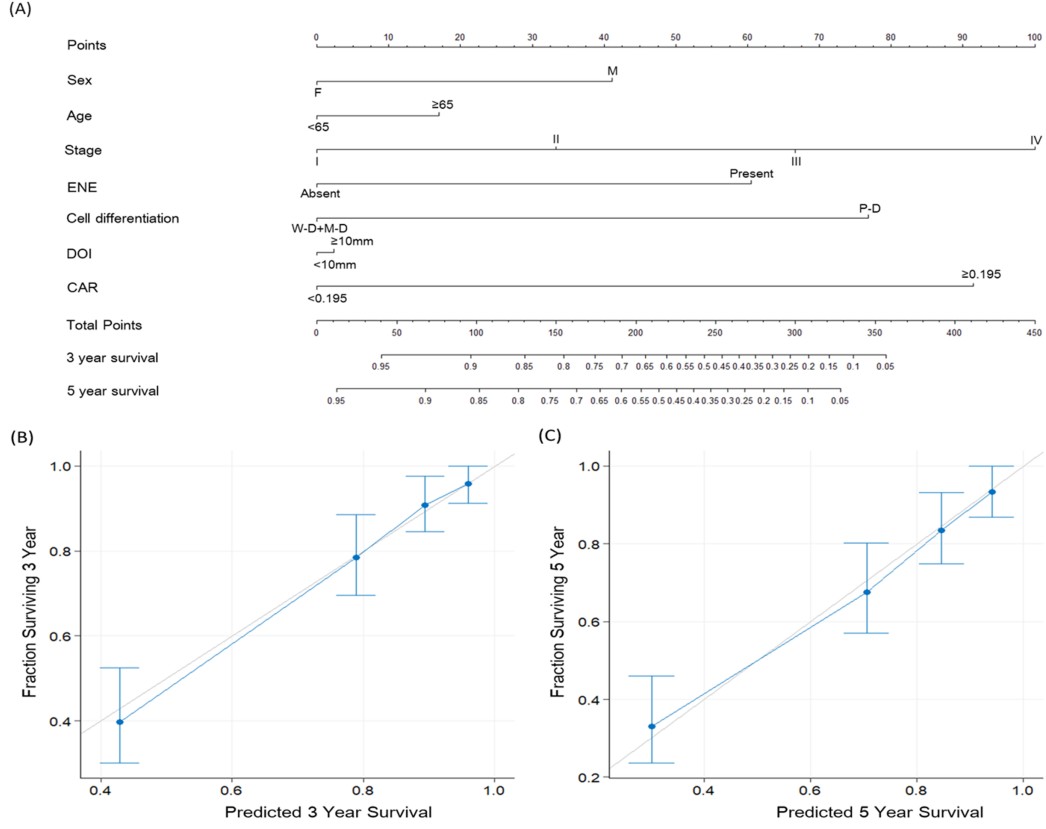

**Figure 5 Nomogram and survival predictions.** (A) Nomogram for OS prediction for patients with OSCC. A line runs vertically from each parameter to the uppermost points. Summing the scores for each parameter provides the total score, which can be translated into survival probabilities along a vertical line drawn from the total score to the 3- and 5-year survival axes. Calibration plots of the nomogram for (B) 3-year and (C) 5-year OS prediction for patients with OSCC. The light gray line indicates perfect prediction, and the blue line indicates the predictive ability of our proposed nomogram. Blue dots with bars represent the performance and 95% confidence interval of the nomogram as applied to the surviving cohorts. Abbreviations: ENE, extracapsular nodal extension; DOI, depth of invasion; W-D, well differentiated squamous cell carcinoma; M-D, moderately differentiated squamous cell carcinoma; P-D, poorly differentiated squamous cell carcinoma; CAR, C-reactive protein to albumin ratio.

with head and neck cancer (*Takenaka et al., 2018a*; *Yang et al., 2019*). In our executed study, CAR demonstrated a discrimination ability superior to that of other scores based on inflammation, namely CRP, mGPS, NLR, and PLR, in resectable OSCC patients—consistent with findings reported by previous research executed among patients with other types of cancer (*Liu et al., 2015*; *Wei et al., 2015*). *Park, Kim & Kim (2016)* also found that in a long-term follow-up, the AUCs of CAR were consistently higher than the AUCs of the other inflammation-based prognostic scores, and CAR was the only significant prognostic indicator in patients with OSCC after long-term evaluation. This may partly be explained by preoperative CAR being a straightforward ratio with a continuous range of values; by contrast, mGPS comprises dichotomized variables that are inherently qualitative with discontinuous values. Notably, the present study discovered that the AUCs of CAR and mGPS, calculated using the CRP and albumin levels, were higher

than those of NLR and PLR, which are common prognostic indicators in OSCC (*Jariod-Ferrer et al., 2019*). These results suggest that the prognostic predictive ability is higher with the CRP-based prognostic score than with the peripheral blood cell count-based prognostic scores in patients with OSCC; CAR integrates the patterns of systemic inflammation and host nutritional status, thus enabling it to more effectively reveal prognostic outcomes.

In the present study, we assumed the CAR cutoff of 0.195 may be useful for predicting OS and DFS in patients with OSCC, and this index may have a cutoff specific for primary tumors at different sites (*He et al., 2016*; *Kuboki et al., 2019*; *Park, Kim & Kim, 2016*; *Yu et al., 2017*). Among all studies on head neck cancers, *Wang et al. (2019)* reported the highest cutoff CAR value of 0.525 as a prognostic indicator in patients with OSCC; their study, however, included high proportions of patients aged >60 years and with advanced disease stage. A recent study on CAR's prognostic value in enrolled patients with advanced hypopharyngeal cancer, two-thirds of whom were aged >65 years, also suggested a relatively high CAR cutoff of 0.32 to be useful for predicting prognosis (*Kuboki et al., 2019*). *Yu et al. (2017)* found a CAR of 0.047 for predicting the laryngeal cancer prognosis; they enrolled a large proportion of patients with early-stage laryngeal cancer and relatively young population. On the basis of these study results, the primary tumor site, cancer stage, age distribution of the study cohort, and albumin physiological decrease with aging may all account for the different cutoffs of CAR in head and neck cancer patients.

This study's strengths are twofold: (1) we included a relatively large cohort of patients with resectable OSCC and long follow-up period. (2) Preoperative CAR is a highly accessible biological marker that could be applied in daily clinical practice because CAR measurement is easy and noninvasive and causes no additional burden to the patients. However, this study also has two limitations: (1) The retrospective study design has its inherent limitations. Moreover, studies performed at a single center may lead to selection bias. (2) This study investigated preoperative CAR, a measure that may be affected by such factors as undetected infection and cancer-related inflammation. This warrants further investigation of mechanisms specifically underlying the prognostic value of CAR. Finally, although our results suggest that CAR has prognostic value in patients with resectable OSCC, large multi-institute prospective studies are warranted.

## CONCLUSION

Preoperative CAR was found to be an independent prognostic factor regarding OS and DFS in OSCC patients treated with curative surgery, and it has superior discrimination ability to other examined prognostic scores based on inflammation. Higher CAR was significantly correlated with a variety of poor prognosticators. The established multivariate nomogram model that incorporated preoperative CAR and clinicopathological factors into the current TNM staging system might strengthen the accuracy of prognostic prediction for OSCC patients. Due to its simplicity and high availability, CAR can be used as an objective and noninvasive biomarker for prognostication of OSCC patients undergoing curative surgery and help clinical physician to recognize patients at high risk.

## ACKNOWLEDGEMENTS

The authors thank all the members of the HIE lab, Chang Gung Memorial Hospital, for their invaluable help.

### Funding

This study was supported by the grant (CMRPG6G0342) from Chang Gung Memorial Hospital, Taiwan. The funders had no role in study design, data collection and analysis, decision to publish, or preparation of the manuscript.

### Grant Disclosures

The following grant information was disclosed by the authors:
Chang Gung Memorial Hospital, Taiwan: CMRPG6G0342.

### Competing Interests

The authors declare that they have no competing interests.

### Author Contributions

- Ku-Hao Fang performed the experiments, prepared figures and/or tables, and approved the final draft.
- Chia-Hsuan Lai analyzed the data, prepared figures and/or tables, and approved the final draft.
- Cheng-Ming Hsu analyzed the data, authored or reviewed drafts of the paper, and approved the final draft.
- Ethan Huang analyzed the data, authored or reviewed drafts of the paper, and approved the final draft.
- Ming-Shao Tsai performed the experiments, prepared figures and/or tables, authored or reviewed drafts of the paper, and approved the final draft.
- Geng-He Chang conceived and designed the experiments, authored or reviewed drafts of the paper, and approved the final draft.
- Yi-Chan Lee conceived and designed the experiments, prepared figures and/or tables, authored or reviewed drafts of the paper, and approved the final draft.
- Yao-Te Tsai conceived and designed the experiments, performed the experiments, prepared figures and/or tables, authored or reviewed drafts of the paper, and approved the final draft.

### Human Ethics

The following information was supplied relating to ethical approvals (i.e., approving body and any reference numbers):

This study obtained approval from the institutional review board of Chang Gung Memorial Hospital's Institutional Review Board (201901573B0).

## Data Availability

The raw measurements are available in a Supplemental File.

## Supplemental Information

Supplemental information for this article can be found online at http://dx.doi.org/10.7717/peerj.9361#supplemental-information.

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
