# Peer review of "A retrospective study on the prognostic value of preoperative C-reactive protein to albumin ratio in patients with oral cavity squamous cell carcinoma"

_PeerJ, doi:10.7717/peerj.9361_

## Round 0.1 · original submission · Major Revisions

Your manuscript has been reviewed and requires modifications prior to making a decision. The comments of the reviewers are included at the bottom of this letter. I would request for the manuscript to be revised accordingly, and also like to suggest the following changes:

-Did the authors check the normality assumption of the data before they used the Pearson correlation test? If yes, they should mention this issue in the statistical analysis section, and provide the name of the normality test which they preferred to use.

- In Table 2, the authors used independent samples t-test but they did not mention that they check the normality assumption of the data. Please check the normality assumption and give the name of the normality test.

·

Basic reporting

No comment.

Experimental design

Please have a look at Major concerns: 1 to 2.

Validity of the findings

No comment.

Additional comments

Oral cancers are the sixth most frequent cancer with a high mortality rate. Oral squamous cell carcinoma (OSCC) accounts for more than 90% of all oral cancers. Increasing incidence of OSCC of the oral cavity and oropharynx is reported in young adults who habit of alcohol, cigarette and betel. People’s cigarette smoking, betel quid chewing, and alcohol consumption not only cause field cancerization but are also correlated significantly with systemic inflammation. C-reactive protein (CRP) is a commonly used marker of systemic inflammation, routinely measured in serum blood samples and easy to detect. It is a good idea to investigate the correlation of CRP with OSCC prognosis, and it might provide new ideas for the diagnosis and treatment of OSCC. In this manuscript, Dr. Fang and colleagues looked at the role of the C-reactive protein-to-albumin ratio (CAR) in diagnosis and prognosis of OSCC. The study provides the conclusion, which showed that preoperative CAR could constitute an independent prognostic indicator for OS and DFS prediction in OSCC patients treated with curative surgery. Although, the current study is interesting and well but there are some major concerns that need to be addressed.

Major concerns:
1. Although, there were some reports [1-3] have pointed out the prognostic value of C-reactive protein/Albumin Ratio (CAR) in oral squamous cell carcinoma (OSCC), this study used more complete and larger sample size clinical data. However, the factors affecting the survival of patients are complex, and the patient's physical state and treatment strategy are also different in different clinical stages. Therefore, I suggest that the authors can further analyze the diagnostic efficiency separately in early (might Stage I-II) and late stage (might Stage III-IV) of OSCC. In this study, there were 135 patients in the early stage (Stage I-II) and 191 patients in the late stage (Stage III-IV). I believe there should be enough data for relevant analysis, and it will help to increase the application value of this research.
[1] Wang Q, Song X, Zhao Y, et al. Preoperative high c-reactive protein/albumin ratio is a poor prognostic factor of oral squamous cell carcinoma. Future Oncol. 2019;15(19):2277–2286. doi:10.2217/fon-2019-0063
[2] Park HC, Kim MY, Kim CH. C-reactive protein/albumin ratio as prognostic score in oral squamous cell carcinoma. J Korean Assoc Oral Maxillofac Surg. 2016;42(5):243–250. doi:10.5125/jkaoms.2016.42.5.243
[3] Hasegawa T, Saito I, Takeda D, et al. Risk factors associated with postoperative delirium after surgery for oral cancer. J Craniomaxillofac Surg. 2015;43(7):1094–1098. doi:10.1016/j.jcms.2015.06.011
2. C-reactive protein (CRP) is a commonly used marker of systemic inflammation, and there are some studies have showing that CRP level can be used as a prognostic and diagnostic indicator for esophageal squamous cell carcinoma and tongue squamous cell carcinoma [4-5]. Thus, I suggest that the authors should compare the efficacy of CRP level and CAR in predicting the prognosis of OSCC. I believe that it will enhance the significance of this study.
[4] Katsurahara K, Shiozaki A, Fujiwara H, et al. Relationship Between Postoperative CRP and Prognosis in Thoracic Esophageal Squamous Cell Carcinoma. Anticancer Res. 2018;38(11):6513–6518. doi:10.21873/anticanres.13016
[5] Du J, Hu W, Yang C, Wang Y, Wang X, Yang P. C-reactive protein is associated with the development of tongue squamous cell carcinoma. Acta Biochim Biophys Sin (Shanghai). 2018;50(3):238–245. doi:10.1093/abbs/gmy004

Minor comments:
1. “RESULTS – Baseline characteristics” Line 171-172. “Nearly half of the patients were diagnosed as having stage IV disease…” Please add the number and percentage of patients with stage IV in manuscript as same as other characteristics.
2. Figure 1-3. The text in the picture is too small. Please increase the font size.
3. “RESULTS – Association of CAR with survival outcomes” Line 196-215, Figure 3A-B. Please present the median survival time, hazard ratio (HR) and 95% CI of ratio at survival curve.

Reviewer 2 ·

Basic reporting

No comment

Experimental design

No comment

Validity of the findings

No comment

Additional comments

The paper is relevant for the current literature, minor revisions are needed. It is a really good study. The english is correctly written, clear and unambiguous.

Introduction:
It is rigth

Material and methods:
- Study patients
You should add the literatura about the guidlines followed by your department to the complementary treatment after radical surgery.
Do you request any radiological explorations for the follow up?
- Inflammation – based prognostic scores
It is rigth
- Statistical analysis
It is rigth

Results:
- Baseline characteristics
It is rigth
- Inflammation - based prognostic score cutoff and ROC curves
It is rigth
- Association of CAR with clinicopathological charasteristcs
It is rigth
- Association of CAR with survival outcome
It is rigth
- Nomogram models
It is rigth

Discussion:

In the third paragraph you should cited a current study which AUCs based on inflammation are inferior to CAR in OSCC.
For Example:
Jariod-Ferrer ÚM, Arbones-Mainar JM, Gavin-Clavero MA, Simón-Sanz MV, Moral-Saez I, Cisneros-Gimeno AI, Martinez-Trufero J. Are Comorbidities Associated With Overall Survival in Patients With Oral Squamous Cell Carcinoma? J Oral Maxillofac Surg. 2019 Sep;77(9):1906-1914.

Conclusions:
It is rigth

Reviewer 3 ·

Basic reporting

the english language is clear.
the author should revise the use of comma. ( before AND comma is not necessary)

Experimental design

The reasearch is in line with the aims of the journal
the identification of patients afftected by OSCC that need to more aggressive adjuvant strategy is an open issue.
a retrospective study has a lot of bias
It's not clear if the patients gave the IC.

Validity of the findings

In the multivariate analysis the 5yrs OS is atatically associted with pT, pN , G,rading, CAR and NLR.
it's not clear the reason why tha author used in the nomogram characteristics that weren't associated ( at MVA) with OS such as sex, age and stage.
the patients were enrolled from 2008 to 2017, it's not clear why the authors considered 5years OS.

---

## Round 0.2 · accepted · Accept

The authors addressed the reviewers' concerns and substantially improved the content of MS.

So, based on my own assessment as an editor, no further revisions are required and the MS can be accepted in its current form.

·

Basic reporting

No comment.

Experimental design

No comment.

Validity of the findings

No comment.

Additional comments

Oral cancers are the sixth most frequent cancer with a high mortality rate. Oral squamous cell carcinoma (OSCC) accounts for more than 90% of all oral cancers. Increasing incidence of OSCC of the oral cavity and oropharynx is reported in young adults who habit of alcohol, cigarette and betel. People’s cigarette smoking, betel quid chewing, and alcohol consumption not only cause field cancerization but are also correlated significantly with systemic inflammation. C-reactive protein (CRP) is a commonly used marker of systemic inflammation, routinely measured in serum blood samples and easy to detect. It is a good idea to investigate the correlation of CRP with OSCC prognosis, and it might provide new ideas for the diagnosis and treatment of OSCC. In this manuscript, Dr. Fang and colleagues looked at the role of the C-reactive protein-to-albumin ratio (CAR) in diagnosis and prognosis of OSCC. The study provides the conclusion, which showed that preoperative CAR could constitute an independent prognostic indicator for OS and DFS prediction in OSCC patients treated with curative surgery. The paper is significantly improved and all concerned raised by the reviewer have been addressed. I think it is suitable for publication at this point for this version of revised manuscript.

Reviewer 2 ·

Basic reporting

No comment

Experimental design

No comment

Validity of the findings

No comment

Additional comments

The minnor reviews have been improved. The structure and the design are adecuate. The conclusions are clear and the results are important for the literature.